# Early Postoperative Pneumothorax Might Not Be ‘True’ Recurrence

**DOI:** 10.3390/jcm10235687

**Published:** 2021-12-02

**Authors:** Wongi Woo, Chong Hoon Kim, Bong Jun Kim, Seung Hwan Song, Duk Hwan Moon, Du-Young Kang, Sungsoo Lee

**Affiliations:** 1Department of Thoracic and Cardiovascular Surgery, Gangnam Severance Hospital, Yonsei University College of Medicine, Seoul 06273, Korea; woopendo@gmail.com (W.W.); kjhightech86@yuhs.ac (C.H.K.); claris@yuhs.ac (B.J.K.); pupupuck@yuhs.ac (D.H.M.); 2Department of Thoracic and Cardiovascular Surgery, Sanggye Paik Hospital, Inje University College of Medicine, Seoul 01757, Korea; lenierfree@naver.com; 3Department of Thoracic and Cardiovascular Surgery, Kangbuk Samsung Hospital, Sungkyunkwan University College of Medicine, Seoul 03181, Korea

**Keywords:** pneumothorax, recurrence of pneumothorax, bullectomy, VATS

## Abstract

Objectives: To date, there is no consensual definition of what constitutes a postoperative recurrence of primary spontaneous pneumothorax (PSP), despite there being many studies reporting a high incidence of recurrence. This study aims to describe the long-term recurrence rates of pneumothorax and to suggest a possible way to differentiate recurrence events based on temporal patterns. Methods: This single-center study retrospectively evaluated all postoperative recurrence of PSP from January 2007 to May 2019. Patients’ demographics, history of pneumothorax, radiologic data, surgical technique, and the time between operation and recurrence were analyzed. Univariate and multivariable analyses were conducted to find potential risk factors related to long-term recurrence. Results: Of the 77 postoperative recurrent cases of pneumothorax, 21 (27.2%) occurred within 30 days after surgery and, thus, were classified as early recurrences (ER), while the remaining cases were classified as late recurrences (LR). There was no difference in preoperative variables between the two groups. However, the rate of incidence of second recurrence (SR), which represented a long-term prognosis, was significantly higher in the LR group (28.6% vs. 4.8%, *p* = 0.030). On univariate and multivariable analyses, late recurrence was the only significant factor predicting later recurrence events. Conclusion: Postoperative recurrence (PoR) within 30 days had a lower SR rate. Therefore, it might not be a ‘true’ postoperative recurrence with a favorable prognosis. Further studies investigating postoperative recurrence based on temporal patterns would be warranted to improve the classification of PoR.

## 1. Introduction

With the development of thoracoscopic surgery, it has become more efficient to treat patients with primary spontaneous pneumothorax (PSP). Many studies have shown the benefits of video-assisted thoracoscopic surgery (VATS) compared with open thoracotomy in terms of invasiveness, postoperative bleeding, drainage amount from the chest tube, and hospital stay. However, the recurrence rate seems to be higher after VATS than after open thoracotomy. Many studies have reported a high recurrence rate after VATS, ranging from 16.3 to 24.1% [1,2,3].

Therefore, there has been a considerable amount of research to determine the potential risk factors predicting the postoperative recurrence (PoR) of PSP. Many surgical procedures, such as mechanical pleurodesis, pleural abrasion, pleurectomy, and coverage materials, have been introduced to reduce postoperative PSP [1,3,4,5,6,7,8]. However, to the best of our knowledge, a study about patients with PoR and their long-term recurrence-free survival has not yet been investigated.

In addition, there is currently no clearly established and consensual definition of what constitutes a recurrence of pneumothorax after surgery [8,9]. When previous studies classified recurrence, they lacked a clear explanation about the timing, or time pattern, of PoR. Therefore, our study tried to suggest a new classification system of PoR and described the time pattern of PoR. In addition, we evaluated the risk factors related to second recurrence (SR) to obtain a better understanding of the long-term prognosis of PoR of PSP.

## 2. Methods

We retrospectively reviewed patients who experienced PoR of PSP from January 2007 to May 2019 in our institution. Patients with underlying lung diseases, secondary pneumothorax, catamenial pneumothorax, Marfan syndrome, or iatrogenic pneumothorax were excluded. This study received approval from the Institutional Review Board of Gangnam Severance Hospital (approval NO. 3-2021-0219). Patients’ consent was waived because this retrospective study did not affect any clinical decisions or treatments. We reviewed patients’ demographic data (age, sex, height, weight, smoking status, prior history of pneumothorax, etc.), radiologic images on computed tomography (pre-existing bullae), and other operative data on electrical medical records (EMR).

### 2.1. Inclusion Criteria and Definition of Terms

After having surgery for pneumothorax, patients’ chest X-rays (CXR) were evaluated to determine whether the operated lung was fully expanded. If there was any unexpanded space after the chest-tube removal, serial CXRs were undertaken for 24–48 h. When the spaces did not increase and patients had no complaints of symptoms, they were regarded as dead space from the VATS bullectomy. All patients were followed up at the outpatient clinic within 5–8 days with an evaluation of CXRs. This study included all of the PoR cases that had no change in CXRs at the first outpatient visit compared with those taken at the time of discharge.

PoR of PSP is defined as the second ipsilateral episode of PSP after an operation. Patients were divided into two groups: the early recurrence (ER) group, which included patients who experienced PoR within 30 days after surgery, and the late recurrence (LR) group, which included patients who experienced PoR after 30 days. Second recurrence (SR) is another episode of pneumothorax after receiving treatment for PoR.

### 2.2. Surgical Procedure and Pleurodesis

All surgical procedures were conducted via VATS under general anesthesia with single-lung ventilation. Bullae were identified through a thoracoscope and resected using endoscopic staplers. Additional procedures, such as polyglycolic acid (PGA) sheet coverage, fibrin glue injection, mechanical pleurodesis, or pleural abrasion, were conducted based on the surgeons’ preferences. From 2016, intraoperative mechanical pleurodesis was performed only in limited cases due to concerns about postoperative bleeding. Additionally, if patients had prolonged (over 48 h) air leakage, they were recommended to have chemical pleurodesis with Viscum (ABNOVA Viscum^®^ Fraxini Injection, manufactured by ABNOVA GmbH, Pforzheim, Germany).

### 2.3. Statistical Analysis

All measured data were analyzed by using SAS version 9.4 (SAS Institute, Cary, NC, USA) and R package version 3.6.0. The two groups were statistically analyzed using a Chi-square test, Fischer’s exact test, or an independent t-test. In addition, SR-free survival and SR outcome were described using a Kaplan–Meier curve and a log-rank test. The optimal cut-off point of SR-free survival was calculated using the Contal method, O’Quigley’s method, and the K-Adaptive Partitioning method. A *p*-value < 0.05 was defined as statistically significant.

## 3. Results

During the study period, a total of 811 patients had 865 VATS bullectomies due to PSP (*n* = 841), secondary pneumothorax (*n* = 10), catamenial pneumothorax(*n* = 7), and pneumothorax related to other genetic diseases (*n* = 7). Of the 841 PSP cases, there were 77 (9.2%) PoR events after surgery (74 patients, but three patients had more than one bilateral recurrent pneumothorax occurrence that were counted as different events). Figure 1 presents the frequency of PoR based on postoperative days. In total, we analyzed 77 (9.2%) cases of recurrence; 21 were ER, and 56 were LR.

Patients in the two groups were followed up for a similar period (mean follow-up duration: ER, 30.1 ± 36.3 months; LR, 37.3 ± 29.1 months; *p* = 0.372). There were no statistically significant differences in terms of demographic factors (age, sex, smoking history, body-mass index, and serum albumin level), previous pneumothorax history (total lung collapse at the first pneumothorax and prior contralateral pneumothorax before surgery), radiographic finding (the existence of bullae on a CT scan), and surgical procedures (multiple wedge resection, PGA sheet usage, pleural abrasion, and mechanical pleurodesis). However, the ER group had a significantly lower SR rate than the LR group (4.8% vs. 28.6%, *p* = 0.030) (Table 1 and Table 2).

Intraoperative mechanical pleurodesis was performed in 317 (36.6%) cases to lower the recurrence rate during the entire study period. In this study population, 26 (33.8%) cases received mechanical pleurodesis during the first VATS bullectomy. After the recurrence, the proportion of patients receiving pleurodesis during the second surgery was similar between the ER and LR groups (ER: 3/6(50.0%), LR: 6/19(31.6%), *p* = 0.672). Later, eight patients underwent a third surgery for SR, and of them, three patients (ER: 0/1 (0.0%), LR: 3/7(42.9%), *p* = 0.408) received mechanical pleurodesis.

Factors associated with SR were evaluated using univariate analyses. These factors included age, bullae on a CT scan, a prior contralateral pneumothorax event before surgery, body-mass index, multiple wedge resection, pleurodesis, and late recurrence. In univariate analyses, body-mass index and late recurrence with *p*-values of less than 0.1 were included in the multivariable analysis. The multivariable Cox proportional hazard analysis showed only late recurrence as a significant factor for SR rate (HR 7.957 (95% CI 1.043–60.673), *p* = 0.045) (Table 3). Moreover, we analyzed the data to find the optimal cut-off point of SR-free survival; it was determined to be 30 days (*p* = 0.0206) based on the Contal method and 23 days (*p* = 0.0062) based on O’Quigley’s method. Figure 2 and Figure 3 show the Kaplan–Meier SR-free survival curves with two different cut-off values.

## 4. Discussion

This study analyzed 77 cases of PoR and explained that the temporal patterns of recurrence might be predictive of SR. As we described in Figure 1, 27.3% (21/77) of the cases of PoR occurred within 30 days after surgery. This result is similar to the findings of Brophy et al., which showed that 26.3% (10/38) of PoR occurred within 30 days after surgery [8]. Moreover, patients in the ER group showed a better prognosis in terms of developing recurrent pneumothorax down the road than those in the LR group, and the period between surgery and recurrence played a significant prognostic factor.

To date, there has not been a clear consensus on the definition of postoperative recurrence of PSP. Brophy et al. described a “true” postoperative recurrence as an episode that occurs at least 15 days after surgery and classified repetitive pneumothorax within 15 days as a “prolonged air leak” [8]. Onuki et al. defined recurrence as an episode occurring at least 30 days after surgery and thought that recurrent events within the first 30 days after surgery were part of the staple-healing process [10]. On the other hand, Hsu et al. considered any recurrence at any time after an operation to be PoR [7], while Kim et al. considered a one-year mark as the cutoff to define early recurrence [11]. However, to the best of our knowledge, no study has evaluated the relationship between different clinical prognoses and the temporal patterns of recurrence after surgery. In our study, the ER group showed a much better clinical prognosis with respect to SR; accordingly, early repetitive pneumothorax events that occur within 30 days should be considered to be part of the healing process or as a result of air leakage, exudation, or lung detachment. Therefore, a ‘true’ PoR of pneumothorax could be defined as an episode occurring at least 30 days after an operation due to its distinct clinical manifestations.

The mechanism of PSP has generally been suggested to be associated with the rupture of a subpleural bleb or bullae [12,13]. Although not sufficient, a few studies suggested the neo-bullae genesis theory to explain PoR of PSP [10,12]. Onuki et al. evaluated nine PoR cases (mean follow-up duration, 869 ± 542 days) with CT, and seven of them had different types of neo-genetic bullae, which were found within 3 cm of the staple lines [10]. If we consider the time it takes for the neo-bullae to be formed, LR is thought to be more related to neo-bullae genesis. On the other hand, the cause of ER can be explained as a minor air leakage due to the healing process of surgical stapling sites [14]. Therefore, when PoR occurs in PSP patients, we might apply different treatment strategies depending on the timing of the PoR in the postoperative period. Conservative treatments, such as oxygen therapy, needle aspiration, or closed thoracostomy, can be applicable for ER patients. Nonetheless, further studies comparing the various treatments for both groups are warranted.

Meanwhile, many studies tried to find risk factors related to PoR of PSP in terms of demographic variables (age, gender, and smoking) [2,15,16], pneumothorax history (large size of the first episode) [17], radiological findings (presence of a bleb or bullae on CT and diaphragmatic tenting) [18,19], and surgical techniques (mechanical or chemical pleurodesis, vicryl mesh, and PGA coverage) [2,4,7]. Recently, several meta-analyses reported the benefits of PGA coverage [20] and mechanical and chemical pleurodesis [21]. However, there was not much attention on the timing of PoR, and there was no risk analysis for SR. Our study showed that the SR rate was much higher in the LR group (28.6%, *p* < 0.05) compared with the ER group (4.8%). Moreover, according to our multivariable analysis, the temporal pattern—or the timing of recurrence—after surgery is an independent factor predicting SR. The cut-off values for SR were 30 days based on the Contal method and O’Quigley’s method and 23 days based on the K-adaptive partitioning method. Due to limited evidence and inadequate understanding, at this time, of the natural processes of PoR, further studies considering the temporal aspects of PoR are warranted.

There are several limitations to this study. This is a non-randomized, retrospective, single-center study, and the number of cases is small. Multicenter prospective trials are needed to confirm our results.

In conclusion, patients who experienced PoR of PSP within 30 days showed a better prognosis than patients with late recurrent episodes. Therefore, temporal aspects should be considered in defining a ‘true’ PoR of PSP. Further randomized and prospective studies would be necessary to better understand and confirm the temporal effect on PoR.

## Figures and Tables

**Figure 1 jcm-10-05687-f001:**
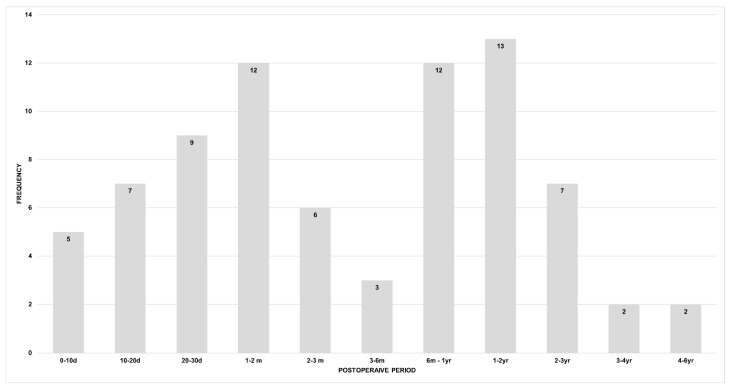
Frequency of postoperative recurrence (d: day; m: month; yr: year).

**Figure 2 jcm-10-05687-f002:**
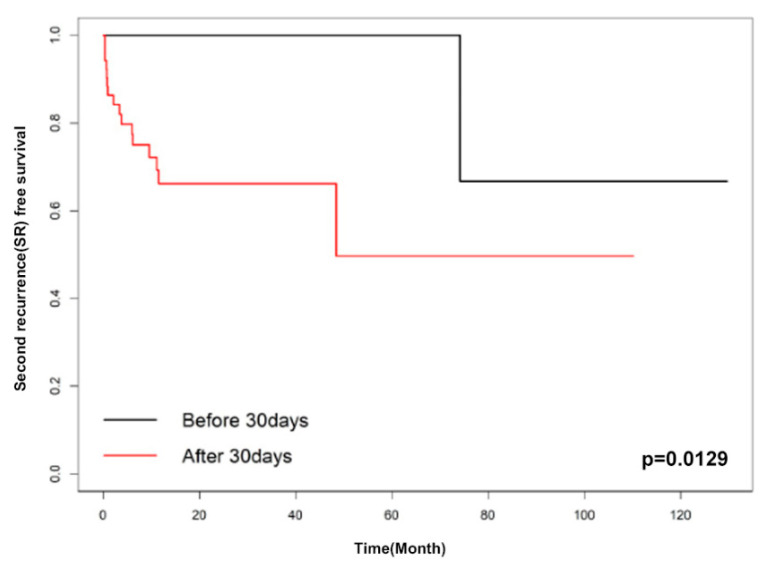
Kaplan–Meier curve of early recurrence and late recurrence (cut-off: 30 days).

**Figure 3 jcm-10-05687-f003:**
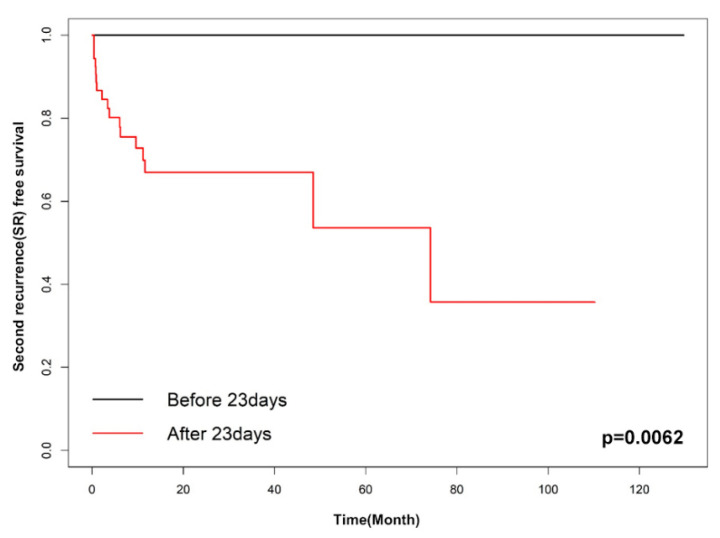
Kaplan–Meier curve of early recurrence and late recurrence (cut-off: 23 days).

**Table 1 jcm-10-05687-t001:** Comparison of baseline characteristics between the early recurrence and late recurrence groups after surgery.

Variables	ER (*n* = 21)	LR (*n* = 56)	*p*-Value
Age, mean ± SD, years	23.2 ± 10.9	25.0 ± 15.7	0.636
Gender, *N* (%)			0.435
Male	20 (95.2)	59 (87.5)	
Female	1 (4.8)	7 (12.5)	
Smoking history, *N* (%)			0.676
No	18 (85.7)	51 (91.1)	
Yes	3 (14.3)	5 (8.9)	
BMI, mean ± SD, kg/m^2^	19.48 ± 2.54	19.39 ± 2.98	0.910
Serum albumin, mean ± SD, mg/dL	4.83 ± 0.32	4.67 ± 0.34	0.079
Total lung collapse episode, *N* (%)			>0.999
No	21 (100)	54 (96.4)	
Yes	0 (0)	2 (3.6)	
Postoperative interval of recurrence, mean ± SD, days	16.4 ± 7.3	416 ± 439	<0.001 **
Definite bullae on CT, *N* (%)			0.298
No	2 (9.5)	2 (3.6)	
Yes	19 (90.5)	54 (96.4)	
History of contralateral pneumothorax, *N* (%)			0.537
No	16 (76.2)	46 (82.1)	
Yes	5 (23.8)	10 (17.9)	
Multiple wedge resection, *N* (%)			0.511
No	16 (76.2)	47 (83.9)	
Yes	5 (23.8)	9 (16.1)	
PGA coverage, *N* (%)			>0.999
No	2 (9.5)	6 (10.7)	
Yes	19 (90.5)	50 (89.3)	
Pleural abrasion, *N* (%)			0.291
No	16 (76.2)	49 (87.5)	
Yes	5 (23.8)	7 (12.5)	
Mechanical pleurodesis, *N* (%)			>0.999
No	14 (66.7)	37 (66.1)	
Yes	7 (33.3)	19 (33.9)	
Chest tube indwelling time, mean ± SD, days	2.95 ± 1.72	2.84 ± 1.41	0.769

** *p* < 0.001; ER: Early recurrence group; LR: Late recurrence group; SD: standard deviation; BMI: Body-mass index; CT: Computed tomography; PGA: polyglycolic acid.

**Table 2 jcm-10-05687-t002:** Postoperative recurrence (PoR) management between the two groups.

Variables	ER Group (*n* = 21)	LR Group (*n* = 56)	*p*-Value
Admission, *N* (%)			0.430
No	1 (4.8)	8 (14.3)	
Yes	20 (95.2)	48 (85.7)	
Treatment methods, *N* (%)			0.351
Conservative care	2 (9.5)	10 (17.9)	
Chest-tube insertion	13 (61.9)	27 (48.2)	
Surgery	6 (28.6)	19 (33.9)	
Hospital stay, mean ± SD, days	4.10 ± 2.15	3.50 ± 2.95	0.415
Second Recurrence, *N* (%)			0.030 *
No	20 (95.2)	40 (71.4)	
Yes	1 (4.8)	16 (28.6)	
Follow-up period, mean ± SD, months	30.1 ± 36.3	37.3 ± 29.1	0.372

* *p* < 0.05; ER: Early recurrence; LR: Late recurrence; SD: Standard deviation.

**Table 3 jcm-10-05687-t003:** Risk factor analysis for second recurrence (SR).

Variables	Univariable Analysis		Multivariable Analysis	
	HR (95%CI)	*p*-Value	HR (95%CI)	*p*-Value
Age				
<20 years	ref			
≥20 years	0.869 (0.244–3.093)	0.828		
Bullae on CT				
No	ref			
Yes	1.901 (0.103–35.091)	0.666		
Contralateral pneumothorax history				
No	ref			
Yes	0.859 (0.246–2.997)	0.812		
Body-mass index	0.868 (0.744–1.013)	0.073	0.891 (0.768–1.034)	0.129
Pleurodesis ^¶^	0.928 (0.342–2.518)	0.883		
Multiple wedge resection				
No	ref			
Yes	1.120 (0.364–3.443)	0.843		
Treatment for PoR				
Others than surgery	ref			
Surgery	0.621 (0.202–1.908)	0.406		
Timing of recurrence				
Early recurrence	ref			
Late recurrence	8.610 (1.130–65.616)	0.038 *	7.957 (1.043–60.673)	0.045 *

* *p* < 0.05; HR: Hazard ratio; CI: Confidence interval; CT: Computed tomography; PoR: Postoperative recurrence.^¶^ Mechanical and/or chemical(viscum) pleurodesis during the admission for the first VATS bullectomy.

## Data Availability

The data underlying this article will be shared on reasonable request to the corresponding authors.

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
