# Peer review of "Early Postoperative Pneumothorax Might Not Be ‘True’ Recurrence"

_jcm, 2021, doi:10.3390/jcm10235687_

Round 1
Reviewer 1 Report
Woo and coworkers performed a retrospective study on definition of PSP recurrence. This is of interest as clear characterization of recurrence after PSP surgery is missing in literature. Therefore, this work could help to get consensus in this field.
But some important information I would suggest the authors to add in their study:
- To get an understanding of recurrence rate please report all patients undergoing surgery because of PSP in the study period 2007 – 2019 and comment on pleurodesis in the whole population.
- As pleurodesis is an important factor to avoid recurrence please add this in your risk factor analysis.
- Please describe type of pleurodesis in second and third surgery.
Reviewer 2 Report
Thank you to the Editor for give me the opportunity to review this manuscript.
This paper is a clinical record about patients that experienced a postoperative recurrence of primary spontaneous pneumothorax.
It is a non-randomized, retrospective, single-center study.
The authors observed that patients with postoperative recurrence events within 30 days ("early recurrence") had a lower second recurrence rate compared to the patients that experienced it later ("late recurrence"). They concluded that the "early recurrence" might not be a "true" postoperative recurrence.
In my opinion the observation is potentially interesting but the results are frail: the number of patients is small and the reasoning is not enough to bring to the conclusions of the authors.
Reviewer 3 Report
This is a retrospective study analysing factors associated with pnuemothorax recurrence post VATs surgery in patients with primary spontaneous pneumothorax. I have several comments
- A key missing piece of data is the denominator - how many patients were operated on in this period and how many did NOT have a recurrence? This is vital for external validity
- the exact methodology by which the authors excluded ongoing post operative pneumothorax needs to be very clear. I could not see any data presented on whether cxrs post surgery were required to show full lung expansion for inclusion in this study. Much more detail is required on this aspect
- I would be interested in understanding why authors divided in to recurrence at < and > 30 days? Was this pre-hoc or determined on the data?
- I think the conclusions are suitably cautious given the limitations of the data - it would be good to speculate on what next study is now required to prove their theory
Round 2
Reviewer 3 Report
Many thanks for the changes to the mansucript - no further comments